# Increased Alveolar Heparan Sulphate and Reduced Pulmonary Surfactant Amount and Function in the Mucopolysaccharidosis IIIA Mouse

**DOI:** 10.3390/cells10040849

**Published:** 2021-04-08

**Authors:** Tamara L. Paget, Emma J. Parkinson-Lawrence, Paul J. Trim, Chiara Autilio, Madhuriben H. Panchal, Grielof Koster, Mercedes Echaide, Marten F. Snel, Anthony D. Postle, Janna L. Morrison, Jésus Pérez-Gil, Sandra Orgeig

**Affiliations:** 1Mechanisms in Cell Biology and Disease Group, UniSA Clinical and Health Sciences, University of South Australia, Adelaide, SA 5000, Australia; tamara.paget@mymail.unisa.edu.au (T.L.P.); emma.parkinson-lawrence@unisa.edu.au (E.J.P.-L.); 2Proteomics, Metabolomics and MS-Imaging Core Facility, South Australian Health and Medical Research Institute, Adelaide, SA 5000, Australia; Paul.Trim@sahmri.com (P.J.T.); Marten.Snel@sahmri.com (M.F.S.); 3Department of Biochemistry, Faculty of Biology and Research Institute Hospital 12 de Octubre (Imas12), Complutense University, 28003 Madrid, Spain; cautilio@ucm.es (C.A.); mechaide@pdi.ucm.es (M.E.); jperezgil@bio.ucm.es (J.P.-G.); 4Faculty of Medicine, University of Southampton, Southampton SO16 6YD, UK; M.H.Panchal@soton.ac.uk (M.H.P.); G.Koster@soton.ac.uk (G.K.); A.D.Postle@soton.ac.uk (A.D.P.); 5Early Origins Adult Health Research Group, Health and Biomedical Innovation, UniSA Clinical and Health Sciences, University of South Australia, Adelaide, SA 5000, Australia; janna.morrison@unisa.edu.au

**Keywords:** MPS IIIA, Sanfilippo syndrome, pulmonary surfactant, respiratory dysfunction, heparan sulphate, bronchoalveolar lavage, surfactant proteins, lipids, captive bubble surfactometry, surface activity

## Abstract

Mucopolysaccharidosis IIIA (MPS IIIA) is a lysosomal storage disease with significant neurological and skeletal pathologies. Respiratory dysfunction is a secondary pathology contributing to mortality in MPS IIIA patients. Pulmonary surfactant is crucial to optimal lung function and has not been investigated in MPS IIIA. We measured heparan sulphate (HS), lipids and surfactant proteins (SP) in pulmonary tissue and bronchoalveolar lavage fluid (BALF), and surfactant activity in healthy and diseased mice (20 weeks of age). Heparan sulphate, ganglioside GM3 and bis(monoacylglycero)phosphate (BMP) were increased in MPS IIIA lung tissue. There was an increase in HS and a decrease in BMP and cholesteryl esters (CE) in MPS IIIA BALF. Phospholipid composition remained unchanged, but BALF total phospholipids were reduced (49.70%) in MPS IIIA. There was a reduction in SP-A, -C and -D mRNA, SP-D protein in tissue and SP-A, -C and -D protein in BALF of MPS IIIA mice. Captive bubble surfactometry showed an increase in minimum and maximum surface tension and percent surface area compression, as well as a higher compressibility and hysteresis in MPS IIIA surfactant upon dynamic cycling. Collectively these biochemical and biophysical changes in alveolar surfactant are likely to be detrimental to lung function in MPS IIIA.

## 1. Introduction

Mucopolysaccharidosis type IIIA (MPS IIIA, Sanfilippo syndrome) is one of a group of up to 70 devastating lysosomal storage diseases (LSD) [1,2,3], which occur as a result of dysfunctional lysosomes. MPS IIIA is caused by mutations in the lysosomal hydrolase gene, N-sulfoglucosamine sulfohydrolase (SGSH), which produces sulphamidase that normally catalyses the degradation of the glycosaminoglycan (GAG), heparan sulphate (HS), to monosaccharides and inorganic sulphate for recycling in the cell [4]. Sulphamidase deficiency causes partially degraded HS and components of the entire catabolic pathway to accumulate in the lysosomal organelles and substrates for hydrolysis by downstream lysosomal enzymes are no longer degraded [5]. Specifically, there is an accumulation of gangliosides GM2 and GM3 and cholesterol affecting neuropathology in the brain [4].

Whilst MPS IIIA has devastating consequences in the brain with severe neurological dysfunction and behavioural disturbances, other symptoms occur in the skeletal, immune, muscular, renal, endocrine and exocrine systems, in addition to disordered sleep patterns and respiratory issues [2,6,7,8]. MPS IIIA is frequently associated with severe respiratory infection in mid to late teenage years with pneumonia being the leading cause of death at the median age of 14.5 years in the severe phenotype [6,9,10].

Primary lung involvement is not normally a symptom reported in the mucopolysaccharidoses at birth. Rather, respiratory symptoms manifest secondary to neurological and skeletal pathologies in the form of severe respiratory obstructions in upper and lower airways in MPS I, II, V and VI patients, and to a lesser extent in MPS IIIA patients [11]. However, respiratory distress was the primary presentation in two recent cases of MPS I [12] and one of MPS IIIA [5] where lung tissue biopsies showed interstitial cells with vacuolated cytoplasm containing a flocculent substrate, which was suggestive of GAG storage [5,12]. Significant pulmonary, and specifically alveolar involvement, have been confirmed in several LSD including Gaucher, Niemann-Pick types A and C2, Sandhoff and Sialidosis diseases [13,14,15,16,17], with not only the accumulation of lipids and other substrates, but a consequential alteration in the production of pulmonary surfactant (PS) leading to respiratory distress and dysfunction. Surfactant is produced and stored in alveolar epithelial type II cells, in special organelles, the lamellar bodies, belonging to the endosome-lysosome system which finally secrete surfactant into the liquid lining of the lung, where it forms a surface-active film at the air-liquid interface. Surfactant serves a dual role: to regulate the interfacial surface tension of the lung during the breathing cycle to reduce the work of breathing and in host defence as part of the innate immune system of the lung, where it protects against pulmonary-acquired pathogens. Therefore, compromise of the PS system may lead to symptoms including recurrent cases of ear, nose and throat infections, which present more frequently with age in the mucopolysaccharidoses [11].

Here, we hypothesised that the accumulation of undegraded heparan sulphate in the alveolar epithelium contributes to altered metabolism, including synthesis, secretion and composition of pulmonary surfactant in MPS IIIA, leading to impaired surface activity. The aim of this study was to describe alterations in pulmonary tissue storage of GAG, lipid and protein composition of lung tissue and pulmonary surfactant, and surfactant activity using a naturally occurring MPS IIIA mouse model. Impaired surfactant metabolism and activity may explain the increased susceptibility of MPS IIIA patients to respiratory infections and other pulmonary insults.

## 2. Materials and Methods

### 2.1. Animals

Wild type, heterozygous and homozygous MPS IIIA (CL57BL/6, D31N aspartate to asparagine mutation) mice were obtained from a breeding colony established and maintained at the Women’s and Children’s Health Network (WCHN) animal facility, Adelaide, South Australia [18]. The mouse model has approximately 3–4% sulphamidase activity in the brain, kidney and liver, and closely mimic the pathological, biochemical and behavioural characteristics observed in humans [18,19]. Mice were kept in a standard 12:12 h light-dark cycle and had free access to pelleted food and water with standard care. All experiments were conducted with ethics approval obtained through the WCHN ethics committee (Ethics #AE1054 and AEC1031/12/18).

### 2.2. Tissue and Lavage (BALF) Collection

Mice were humanely killed with sodium pentobarbitone (lethabarb 20–80 mg/100 g body weight) at approximately 20 weeks of age. Mice (*n* = 30 wild type, 25 heterozygous and 55 MPS IIIA mice) were allocated to either tissue collection or bronchoalveolar lavage fluid (BALF) collection. Lung tissue was collected by dissecting the lungs and snap-freezing and storing at −80 °C before analysis. BALF was collected from lungs in situ via tracheal 103 cannulation. Three separate volumes of ice-cold 0.9% (*w*/*v*) saline were each instilled into and withdrawn from the lungs 3 times to rinse the lungs of pulmonary surfactant. Recovered BALF was placed on ice before centrifuging (150× *g*, 5 min, 4 °C). The supernatant was decanted and stored at −80 °C for later lipid extraction and analysis.

### 2.3. Tissue Homogenisation

Lung tissue samples were prepared for analysis by loading silica-bead tubes with ~50 mg tissue, 1 mL phosphate buffer and protease inhibitor (2.5 µL/50 mg tissue) (Roche Diagnostics, cOmplete protease inhibitor Mini cat#04693159001) before homogenising (Precelly’s 24, Bertin Technology) (6000 rpm, 10 s, room temperature (RT)). Samples were centrifuged (Eppendorf centrifuge 5415R) (500 g, 5 min, 4 °C) to pellet the debris. The supernatant was collected and stored at −80 °C. Total protein quantification in lung homogenates was determined using a micro bicinchoninic acid (BCA) protein assay kit (Thermo Scientific Inc., Melbourne, VIC, Australia Cat# PN 23 227). Bovine serum albumin was used to generate the standard curve.

### 2.4. Sample Preparation and Analysis—Heparan Sulphate

Heparan sulphate in tissue and BALF samples was depolymerised and desulphated to prepare alkylated disaccharides by butanolysis according to published methods [20]. Briefly, BALF (900 µL) samples were freeze dried overnight. 2,2-dimethoxypropane (50 µL) and butanolic HCl (3 M, 1000 µL) were added to BALF and tissue lysate samples, sealed and incubated (100 °C, 2 h). Samples were dried under nitrogen (45 °C, 1 h) before being reconstituted with 200 µL d9 disaccharide internal standard (IS). Solutions were mixed (30 min) on an orbital shaker (Ratek Instruments, Boronia, VIC, Australia) and centrifuged (Biofuge pico centrifuge, Heraeus, Germany) (13,000 rpm, 15 min). Standards were prepared using disaccharide standard at 1, 5, 20, 100, 500, 1000 and 1500 ng. Blanks consisted of MilliQ water and 0.1% Trifluoroacetic acid (TFA). Sample supernatant (80 µL) and disaccharide standards were loaded into a 96-well microtiter plate, sealed with foil and placed at 6 °C in the sample manager for analysis.

Liquid chromatography-tandem mass spectrometry (LC-MS/MS) analysis was performed using an Acquity UPLC (Waters Corporation, Milford, MA, USA) equipped with a 2.1 mm ID × 50 mm BEH C18 (1.7 µm particle) analytical column (P/N. 186002350) and an API 4000 QTrap mass spectrometer (ABSciex, Concord, ON, Canada). A random order run was created on the mass spectrometry software prior to commencing the run. Initial blank and quality control samples were in specific positions in the run. Samples were analysed interspersed every three injections with a blank injection of MilliQ water/0.1% TFA. A binary solvent system was used; solvent A consisting of water with 0.1% formic acid (FA) and solvent B consisting of acetonitrile with 0.1% FA. Sample injection (2 µL) was loaded on the column at a flow rate of 350 µL/min using 99% solvent A. The LC system was conditioned (2 min) prior to the commencement of the analysis. Chromatographic separation was performed using the gradient shown in Table 1.

Data were acquired in multiple reaction monitoring (MRM) mode. In this mode six transitions were monitored for 50 ms each, *m*/*z* 468.245 to 162.077 (butanolysis product), and *m*/*z* 477.300 to 162.077 (d9 Disaccharide IS). The peak areas were calculated using Analyst 1.6.2 (AB/Sciex). Although there were multiple peaks present following butanolysis of HS, the dominant peak was integrated for the analysis. The peaks relating to the chondroitin sulphate/dermatan sulphate transition were generally not detected in samples prepared by butanolysis.

### 2.5. Sample Preparation and Analysis—Gangliosides, Bis(Monoacylglycero)Phosphate (BMP), and Cholesterol

The Folch extraction method was used to prepare samples for analysis [21]. Briefly, BALF samples (90 µL) were freeze dried and reconstituted with MilliQ water (100 µL). Internal standards were added to BALF and tissue samples (100 µg protein) to enable quantification. CHCl_3_/CH_3_OH (2 mL) was added prior to mixing on a tube rotator (40 rpm, 10 min) and standing (RT, 50 min). MilliQ water (400 µL) was added, mixed on a tube rotator (40 rpm, 10 min) and then centrifuged (3500 rpm, 5 min). Resultant layers were transferred as follows: upper aqueous layers to glass tubes for ganglioside extraction procedure and lower hydrophobic phase to two glass tubes for BMP and cholesterol analysis, respectively (~50 µL each).

For ganglioside extraction, C18 (200 mg) columns were primed with 3 × 1 mL CH_3_OH followed by 3 × 1 mL MilliQ water and loaded with sample. Columns were washed with 1 × 1 mL MilliQ water before eluting gangliosides with 3 × 1 mL CH_3_OH into 10 mL glass tubes, followed by brief vacuum drying. Samples were dried under nitrogen (38 °C) before resuspending in 100 µL 10 mM NH_4_COOH in CH_3_OH and centrifuging (3500 rpm, 5 min). Supernatant (80 µL) was transferred into a 96-well microtitre plate for analysis.

For BMP extraction, samples were dried under nitrogen at 38 °C, resuspended in 200 µL 10 mM NH_4_COOH in CH_3_OH and vortexed briefly before centrifuging (3500 rpm, 5 min). The supernatant (~25 µL/sample) was transferred to a 96-well microtiter plate for analysis.

For the cholesterol esterification reaction, samples were dried under nitrogen at 38 °C. Anhydrous acetyl chloride/CHCl_3_ (1:5 *v*:*v*) (100 µL) was added, the vial sealed, vortexed (1 min) and allowed to stand (RT, 1 h). Samples were dried under nitrogen at 38 °C, reconstituted in 200 µL 100% CH_3_OH/10 mM NH_4_COOH and centrifuged (3500 rpm, 5 min). Supernatant (~80 µL) was transferred into a 96 well microtitre plate for analysis.

Liquid chromatography-tandem mass spectrometry (LC-MS/MS) analysis was performed using a Shimadzu LC-20AD binary pump system combined with an ABSciex API 4000 QTrap mass spectrometer equipped with Analyst software (Version 1.4.2) and a turbo-ionspray source. Liquid chromatography separation was achieved by injecting samples (20 μL) onto a 3 µm Alltima C18 column (50 × 2.1 mm) at 200 µL. The HPLC gradient program began with 70% mobile phase A (30% tetrahydrofuran/20% CH_3_OH/50% 5 mM NH_4_COOH in H_2_O) followed by a linear ramp (0.01–7.0 min) to 100% mobile phase B (70% tetrahydrofuran/20% CH_3_OH/10% 5 mM NH_4_COOH in H_2_O) and maintained for 3 min. Re-equilibration at 70% mobile phase A was performed for 3 min prior to the next injection. A Valco 10-port post column valve diverted column flow to waste for the first 1.7 min.

Mass spectrometric analysis of gangliosides was performed in negative ion multiple reaction monitoring mode using an ionspray temperature of 200 °C and voltage of −5000 V. Nitrogen was used as the collision gas at a pressure of 2 × 10^−5^ Torr. Concentrations of each molecular species were calculated by relating the peak areas of each species to the peak area of the corresponding internal standard using Analyst 1.6.2 software.

Mass spectrometric analysis of cholesterol and BMP was performed in positive ion multiple reaction monitoring mode using an ionspray temperature of 200 °C and voltage of 5000 V. Nitrogen was used as the collision gas at a pressure of 2 × 10^−5^ Torr. Relative cholesterol levels were determined by relating the peak area of C_2_ cholesterol to the peak area of C17 cholesterol internal standard. BMP molecular species were differentiated from phosphatidylglycerol by their molecular ion fragmentation in positive mode. This allowed identification of both acyl carbon chains for BMP. Concentrations of each molecular species were calculated by relating the peak areas of each species to the peak area of the corresponding internal standard using Analyst 1.6.2 software.

### 2.6. Tissue and Lavage Collection for Total Phospholipid Pool Analysis

Control and MPS IIIA mice (as detailed in Section 2.1) used in the phospholipid pool analysis were collected as part of a larger cohort designed to measure a phosphatidylcholine kinetics time course following intraperitoneal injection of D9 choline chloride (Ethics #AE1054). Here, we presented only the data from the 0 h time point, in which mice were not injected with the isotope. Following very deep anaesthesia that was administered via intraperitoneal injection of xylazine (10 mg/kg)/ketamine (100 mg/kg), the chest cavity was opened, blood collected from the heart, and a sample of liver collected. Plasma was separated and both plasma and liver samples were snap frozen for future phospholipid analysis. Both lungs were lavaged via tracheal cannulation and the BALF prepared and frozen as per Section 4.2. Lavage recovery was 84.5 ± 8.8% from *n* = 9 control, 14 MPS IIIA samples, which is similar to that reported previously [22]. The left lung lobe was snap frozen to use for lung tissue phospholipid analysis (*n* = 16 control, 14 MPS IIIA samples) and the right lung lobes were excised and chilled in 0.25 M sucrose to isolate lamellar bodies (no data shown). Samples were transported on dry ice to Southampton, UK, for mass spectrometry analysis.

### 2.7. Sample Preparation and Analysis—Total Phospholipids

Lavaged lung tissue, with 10 µL of the antioxidant butylated hydroxyl toludene (BHT) added, was homogenised in saline using a Heidolph Silent Crusher S prior to storing a −80 °C. Phospholipids were extracted from both lung tissue homogenates and BALF using the method of Bligh and Dyer [23]. BALF samples were thawed and 800 µL of each sample was removed before the addition of 10 µL of BHT. To the 800 µL aliquots each, of tissue homogenate and BALF, 100 µL of an internal standard mixture of synthetic lipids (Table 2) was added. Samples were vortexed prior to lipid extraction using a Freedom EVO 100 robotic liquid handling system (Tecan, Theale, UK). The robot was programmed to add 2 mL methanol, 2 mL dichloromethane (DCM) and 1 mL of distilled water to each sample and mixed well to allow for the formation of a biphasic mixture; the lower DCM layer containing phospholipids and an aqueous upper layer. Samples were centrifuged (3000 rpm, 10 min, 20 °C) and the lipid layer removed by the robot before being dried under a stream of warmed nitrogen gas (Ultravap, Porvair, Norfolk, UK) for 40 min. Samples were stored at −80 °C until they were analysed by mass spectrometry.

Mass spectrometry was performed on phospholipids using electrospray ionisation (ESI) (Waters XEVO TQ-MS). Dried BALF samples were dissolved in 1 mL 66% CH_3_OH, 30% DCM and 4% C_₂_H_₇_NO_₂_ (300 mM in H_2_O). Two hundred µL was used from each sample to prepare a pooled Analytical Quality Control (AQC) sample for each genotype. The sample solution was infused using the loop injection method into the mass spectrometer (HP1100 Agilent HPLC) at 0.005 mL/min using 99% CH_3_OH with 1% acetic acid with a series of multiple injections. Diagnostic MS/MS scans were carried out in positive and negative mode (Table 3) and analysis was conducted as indicated in diagnostic MS/MS scans for head groups (Appendix A). Measuring time for each sample varied between 35 and 90 min with 5 min cleaning injection (60% methanol, 30% dichloromethane and 10% acetic acid) after every sample. The pooled AQC was run with between 5 to 6 samples for each mouse genotype and with each genotype at the start and at the end of each batch. This extended analysis time enabled a comprehensive range of diagnostic MS/MS scans to be performed, detailed below and summarised in Appendix A.

Positive and negative ESI scans were performed for each sample using an injection volume of 50 µL and an analysis time of 13 min, measuring the total number of ionised molecules in the samples. This was a control for the presence of any contaminants and overall signal intensity and were not used for ion peak assignments.Diagnostic MS/MS precursor scans were performed with the second injection (100 µL) and analysis time of 22.40 min to quantify PC. Scans were based on the detection of the major collision-induced dissociation (CID) phosphorylcholine fragment ion, detected at *m*/*z* 184.The third injection (100 µL, analysis 22.40 min) included neutral loss (NL) scans for the DPPC molecule and diagnostic scans for the other major phospholipid classes, P153 for acidic phospholipids, P241 for phosphatidylinositol (PI), NL141 for phosphatidylethanolamine (PE) and NL87 for phosphatidylserine (PS). The final injection (100 µL, analysis 22.40 min) included precursor scans to determine the distributions of fatty acyl moieties. Each of the lipids indicated Appendix A) were detected by the indicated type of scan to provide a comprehensive analysis of unlabelled phospholipids.

For molecular species analysis mass spectrometry provided a generic formula based on the total number of carbon atoms and unsaturated double bonds in each lipid class (Appendix A). Based on previous analyses [24,25] and MS/MS fragmentation in negative ionisation to generate diagnostic fatty acyl residues, column 2 (Appendix A) identifies the most probable molecular structures. Here, designations are given as PXa:y/b:x for each phospholipid class, where X represents the headgroup base (e.g., choline or glycerol), a and b are the number of carbon atoms in the sn-1 and sn-2 fatty acyl residues and y and x are numbers of double bonds.

### 2.8. Captive Bubble Surfactometry (CBS)

BALF precipitant (*n* = 10 control, 6 MPS IIIA) was prepared by ultracentrifugation (Beckman Optima L-90K; SW65Ti rotor; 31,000 rpm; 4 °C; 1 h) and resuspended in ~10 µL supernatant and placed in a bath sonicator for 8 min. PC was assayed in duplicate using previously published enzymatic methods (Spinreact, St. Esteve de Bas, Girona, Spain) [26,27,28] and samples reconstituted with phosphate buffered saline to a PC concentration of 20 mg/mL. For CBS analysis ~200 nL of the surfactant suspension was applied to the surface of a 0.05 mL air bubble created inside a buffer-filled (5 mM Tris, 150 mM NaCl, pH 7 and 5% sucrose) thermostatically controlled (37 °C) chamber. The bubble was compressed and expanded periodically using hydrostatic pressure delivered by a piston. CBS mimics the respiratory cycle changes in alveolar volumes so that surface-active properties under dynamic compression-expansion conditions are measured at the air-liquid interface of the air bubble. Variations in the bubble shape were continuously recorded and analysed using the custom-designed CBS software, facilitating the calculation of volume, area and surface tension at all times [29]. For each sample, CBS measurements were carried out in triplicate at 37 °C according to the following protocol. After initial adsorption for 5 min, the chamber was sealed and the bubble expanded to 0.15 mL. Post-expansion adsorption was measured for another 5 min. Thereafter a single quasi-static compression cycle was completed to determine the bubble size limits for step-wise 20% compressions to a final minimum bubble size of ~15% of maximum. Thereafter, the bubble was cycled continuously for 30 cycles at 30 cycles/min. The changes in bubble shape were monitored continuously and related to changes in surface tension (γ).

### 2.9. Gene Expression Using Quantitative Real-Time PCR

Total RNA was extracted from ~50 mg lung tissue using Qiagen QIAzol Lysis Reagent and Qiagen RNeasy purification columns (Qiagen Pty Ltd., Doncaster, VIC, Australia) according to the manufacturer’s instructions and as previously described [30,31,32]. Total RNA was quantified using spectrophotometric measurements at 260 and 280 nm while the integrity of purified RNA was verified by assessment of the RNA bands run on an agarose gel stained with ethidium bromide.

cDNA was synthesised according to the manufacturer’s guidelines with a Superscript III First Strand Synthesis System (Invitrogen, Carlsbad, CA, USA), using 2 μg total diluted RNA, random hexamers, dNTP, DTT and Superscript III in a final volume of 20 μL. A no template control (NTC) containing no RNA transcript and a no amplification control (NAC) containing no Superscript III were used to check for contamination of reagents or genomic DNA, respectively [33].

Primer sets for target genes (surfactant proteins) and housekeeping genes were designed from the NCBI database (Table 4). All primer pairs were optimised and validated to determine the minimum primer concentration to yield the maximum response.

The geNorm component of the qBase 2.0 relative quantification model (Biogazelle) was used to select the most stable of the genes. Based on this model, ribosomal protein P0 (RpP0), tyrosine 3-monooxygenase/tryptophan 5-Monooxygenase Activation Protein Zeta (YWAHZ) and peptidylprolyl isomerase A (PPIA) were stably expressed across the control and MPS IIIA groups and hence were selected from the candidate reference genes (Table 4) to act as internal standards in the following real time PCR analyses, which followed the MIQE guidelines [32].

Gene expression was determined by qRT-PCR using Fast SYBR Green master mix (Applied Biosystems) in a final volume of 6 μL using a ViiA7 Fast Real-Time PCR machine (Applied Biosystems). Using DataAssist 3.0 (Life Technologies), all surfactant proteins (SP-A, -B, -C and -D) were normalised to the geometric mean of the three most stable housekeeping genes; RpPO, PPIA and YWHAZ. Each well on the qRT-PCR plate contained 1 μL of cDNA, 3 μL FastSYBRGreen Master Mix (2×), and 2 μL of forward and reverse primer (Table 4) mixed with differing amounts of H_2_O depending on the required final primer concentrations. NTCs for each primer set were included on each plate to check for non-specific amplification. The threshold was set within the exponential growth phase of the amplification curve and the corresponding cycle threshold (Ct) values were obtained to quantify each reaction. A standard curve was obtained by plotting Ct values against the log of cDNA template concentrations. Gene expression was determined by qBase 2.0 relative quantification analysis software (Biogazelle) and expressed as mean normalised expression (MNE).

### 2.10. Protein Expression Using Western Blot Analysis

Tissue lysate homogenisation for Western blot analysis was completed as per Section 2.2. Quantitative Western blot analysis for lung tissue and BALF samples were optimised using the methods, principles and guidelines described by Pillai-Kastoori et al. [34]. Optimal loading efficiency for each of the primary antibodies was determined using antibody validation and linear range analysis (Empiria Studio Software, Millennium Science, Mulgrave, VIC, Australia).

Samples for SP-A and -D analyses were reduced prior to sodium dodecyl sulfate polyacrylamide gel electrophoresis (SDS-PAGE) using 4–12% bis-tris gels (Thermo Scientific cat# NW04125BOX). Samples for SP-B were analysed under non-reducing conditions, and SP-C analysis under reducing conditions before SDS-PAGE using 16% tricine gels (Thermo Scientific cat# EC66955BOX). Protein transfer to PVDF membrane was performed using the Iblot2 transfer system (Thermo Scientific, Melbourne, VIC, Australia). Total protein analysis used Revert total protein stain (Millennium Science Cat#LCR-926-11021) and fluorescence imaging (700 nm) Odyssey^®^ CLX, Millennium Science, Melbourne, VIC, Australia) (Kirshner & Gibbs, 2018) prior to blocking in Odyssey^®^ TBS blocking buffer. Membranes were incubated overnight (4 °C, rocking) in primary antibody (SP-A Anti-human SP-A Rabbit polyclonal Thermo Scientific Cat#PA5-76699; SP-B Anti-Mature SP-B Rabbit polyclonal SevenHills Bioreagents Cat#WRAB48604; SP-C Anti-Mature SP-C Rabbit polyclonal SevenHills Bioreagents Cat#WRAB-MSPC; SP-D tissue Anti-mouse SP-D Rabbit polyclonal Chemicon International Cat#AB3434 and SP-D BALF Anti-SP-D Rabbit polyclonal Thermo Scientific Cat#PA5-81476). Membranes were then incubated in secondary antibody for 1 h (IRDye 800CW Donkey anti-rabbit IgG Millennium Science Cat#LCR-925-32213 or IRDye 800CW Goat anti-rabbit IgG Millenium Science Cat#LCR-925-32211). Image analysis was performed using the Odyssey^®^ CLX and Image Studio™ Lite Quantification Software (Millennium Science, Melbourne, VIC, Australia). All samples were measured in triplicate. Post image analysis was performed using Image Studio™ Lite Quantification software. Surfactant protein was normalised to total protein and data were expressed as arbitrary units.

### 2.11. Statistics

Graphical representation and statistical analyses of data were performed using the software package GraphPad Prism 8.3.1 (GraphPad Software, San Diego, CA, USA. www.graphpad.com, accessed on 8 April 2021). As there was negligible HS present in the heterozygous control lung tissue, indicating that there is very little primary storage, the wildtype and heterozygous samples were combined to create a control group and compared with the MPS IIIA samples. Individual results for WT and heterozygous samples are provided in Appendix A. Control and MPS IIIA groups were analysed using the two-tailed Students t-test, unless stated otherwise, and presented as mean ± SD; *p* < 0.05 was considered significant.

## 3. Results

### 3.1. Heparan Sulphate and Lipid Analyses by Liquid Chromatography Tandem Mass Spectrometry

#### 3.1.1. Presence of Heparan Sulphate (HS) in Lung Tissue and Bronchoalveolar Lavage Fluid (BALF)

The amount of HS in lung tissue (3511 ± 584.5 ng/mg protein, *n* =11) and BALF (400.2 ± 98.3 ng/mL BALF, *n* = 13) of MPS IIIA mice was significantly greater when compared to that of wild type (lung tissue: 64.85 ± 9.0 ng/mg protein, *n* = 9; BALF: 14.78 ± 4.0 ng/mL BALF) and heterozygous mice (75.13 ± 25.5 ng/mg protein, *n* = 9; BALF: 18.86 ± 8.2 ng/mL BALF, *n* = 7). As there was negligible difference in HS measures between wild type and heterozygous mice, we combined the HS measures for wild type and heterozygous mice to make a single control group for HS (Figure 1A,B). The data for wild type and heterozygous animals were combined into a single control group for all further analyses. The data (mean ± SD) for the individual and combined groups for all analytes are presented in Appendix A.

#### 3.1.2. Alterations in Ganglioside Concentration in Lung Tissue

Gangliosides are sialic acid-containing glycosphingolipids. GM3 gangliosides contain three monosaccharide groups attached to a ceramide backbone that consists of a sphingoid base and a fatty acid linked by an amide bond. Different species of GM3 are determined by the fatty acid moieties, which are denoted as d18:1/A:x, where A is the number of carbon atoms and x is the number of double bonds in the fatty acid. The concentration of two of the GM3 ganglioside molecular species d18:1/16:0 and d18:1/22:0 were significantly increased in the lung tissue of MPS IIIA mice when compared to controls (Figure 2). There was no significant difference in the remaining GM3 molecular species nor in GM1 ganglioside concentration in lung tissue between control and MPS IIIA mice (Appendix A) and GM2 gangliosides were below the level of quantitation.

#### 3.1.3. Alterations in Bis(Monoacylglycerol)Phosphate (BMP) Concentration in Lung Tissue and BALF

Total BMP measured in lung homogenates of 20-week old MPS IIIA mice showed a significant increase when compared to control mice (Figure 3). Further extrapolation of the data showed a significant increase in nine of sixteen molecular species (Figure 4, Appendix A).

There were no significant changes in the total amount of BMP measured in BALF of MPS IIIA mice when compared to control mice (Appendix A). However, further analysis of the subspecies of BMP showed significant decreases in three of seventeen molecular species examined when compared with control mice (Figure 5, Appendix A).

#### 3.1.4. Alterations in Cholesterol Concentration in Lung Tissue and BALF

Total cholesterol measured in lung homogenates of MPS IIIA mice showed no significant difference when compared with control mice (Appendix A). Total cholesterol measured in BALF of MPS IIIA mice showed a significant decrease when compared with control mice (Figure 6A). The decrease in BALF was further analysed, showing no difference in free cholesterol (Appendix A) and a significant decrease in cholesteryl esters in four species when compared to control mice (Figure 6B).

### 3.2. Phospholipid Composition and Alterations in Total Alveolar Phospholipid Pool Size

Individual phospholipid headgroup components were measured in control and MPS IIIA BALF. The relative percentage of the individual major phospholipids, phosphatidylcholine and phosphatidylglycerol, and the minor phospholipids, phosphatidylserine and sphingomyelin, as a percentage of the total phospholipid pool remained unchanged (Appendix A). There was a small increase in the % phosphatidylethanolamine (Appendix A). Phosphatidylcholine (PC) molecular species composition (% of total PL) was measured in BALF and lung tissue. Phosphatidylcholine molecular species composition remained unchanged in lung tissue (Appendix A); however, in BALF there was an increase and decrease in two different species, respectively (Appendix A). When all individual BALF phospholipid components were summed and expressed as nmol in the total lavage volume, there was a significant decrease of ~50% in the total pool size of alveolar surfactant in MPS IIIA mice (Figure 7).

### 3.3. Surfactant Protein Gene Expression and Protein Analysis

There was a significant but modest decrease in the gene expression of SP-A, -C and -D in MPS IIIA lung tissue when compared to control tissue (Figure 8). A significant but modest decrease in the amount of SP-D protein was seen in MPS IIIA lung tissue but there was no change in the amount of SP-A, -B and -C (Figure 9). SP-A, -C and -D were significantly reduced in MPS IIIA BALF when compared to control BALF (Figure 10).

### 3.4. Captive Bubble Surfactometry

Captive bubble surfactometry was used to measure the rate of film formation upon initial adsorption (Figure 11A) and immediately following a sudden bubble expansion (Figure 11B), each over 5 min at 37 °C. Under these static conditions, surfactant demonstrated similar initial adsorption between control and MPS IIIA samples, lowering equilibrium surface tension (γeq) to ~20 mN/m (Figure 11A). However, surfactant from MPS IIIA animals demonstrated a decreased post expansion adsorption rate compared to surfactant from control mice (Figure 11B). Under dynamic conditions (30 cycles/min) the biophysical behaviour of MPS IIIA surfactant was highly variable (Figure 12) relative to control samples, with the surfactant of many diseased individual animals performing more poorly. Upon averaging the individual variables, it was evident that MPS IIIA surfactant was unable to reach a low surface tension at minimum bubble size (γ_min_) after 20 cycles (Figure 13A). Similarly, the surfactant of MPS IIIA animals also had a higher surface tension at maximum bubble size (γ_max_) during dynamic cycling (Figure 13B). In addition, the change in area required to reach γ_min_ was significantly greater in MPS IIIA animals (Figure 13C). MPS IIIA samples also had higher compressibility (lower slopes) compared to control samples (Figure 13D), as well as an increased hysteresis (Figure 13E).

## 4. Discussion

MPS IIIA, a lysosomal storage disease with predominantly neurological pathology, is characterised by the primary storage of HS, and secondary storage of gangliosides GM2 and GM3 and cholesterol in the brain [4]. Similar changes in the lung have the potential to impair the function of the alveolar epithelium, thereby leading to changes in surfactant lipid and protein metabolism and hence the surface activity of the mixture acting at the alveolar air-liquid interface. We propose that this alveolar pathology may impact the susceptibility of MPS IIIA patients to respiratory infections and other pulmonary insults.

We reported an increase in HS in both lung tissue and alveolar surfactant of MPS IIIA mice compared with unaffected control mice. We have also found increases in ganglioside GM3 and BMP lipid species, decreases in surfactant proteins -A, -C and -D mRNA expression and a decrease in SP-D protein in lung tissue. Analysis of alveolar surfactant has shown a decrease in BMP and cholesterol and decreases in surfactant proteins -A, -C and -D. Furthermore, although there were no major changes in the phospholipid composition of alveolar surfactant, there was a 49.7% decrease in the total phospholipid pool size in the MPS IIIA mouse lung. Even though the experimental phospholipid concentration was kept constant in vitro, MPS IIIA mice also demonstrated reduced surface activity upon dynamic compression-expansion cycling. This is the first study to have demonstrated an impairment of surfactant composition and function in the mucopolysaccharidoses.

### 4.1. Heparan Sulphate Accumulation in Lung Tissue and the Alveolar Compartment

The many roles of HS are well defined in mammalian tissue where it is expressed ubiquitously in the extracellular matrix (ECM) and basement membrane of tissues, and on cell surfaces [36,37]. Heparan sulphate is involved in many physiological processes including cell signalling, cell adhesion, inflammation and immunity [38,39,40]. The role extends to help modulate the homeostasis of the parenchymal tissue of the lung [40,41], acting as a key structural component to maintain the efficient architecture and functioning of the lung; i.e., for alveolar walls to be thin for appropriate gas exchange, firm to prevent collapse of the alveolus, flexible to cope with volume changes during breathing [41] and as a regulator of signalling pathways [42,43]. When the breakdown of HS is compromised, as in MPS IIIA, it pre-empts major disturbances in the coordination of not only effector molecules, including growth factors and signalling factors, but extends to proteases and ECM molecules inherently impacting the structural integrity of the lung [44,45,46]. However, the effects of HS within the alveolar compartment itself have to our knowledge not previously been investigated.

The HS results in this study using butanolysis followed by LC-MS/MS [20] have shown negligible differences in HS lung tissue content between heterozygous and wild type mice, but a significant increase in MPS IIIA mice, which is similar to the results reported for MPS IIIA mice in a previous study [47]. Furthermore, we report for the first time a significant increase in the content of HS in the BALF of MPS IIIA mice when compared with that of either wild type or heterozygous mice. The presence of undegraded HS in secreted alveolar surfactant may alter or impact the distribution and function of the interfacial surfactant film components. It is possible that this excess substrate may have a similar impact as bacterial lipopolysaccharide (LPS), which is a major outer surface membrane component present in almost all Gram-negative bacteria, and that is capable of incorporating into the surfactant monolayer [48]. This destabilises the interfacial film and inactivates surfactant, therefore disrupting the ability of surfactant to regulate alveolar surface tension [48,49]. Alternatively, it may be possible that the anionic character of the HS polymer may affect the organisation of surfactant, for example by interacting with the cationic hydrophobic proteins, potentially limiting their availability to carry out their biophysical function. Certainly, our surface activity results support the suggestion that the accumulation of alveolar HS affects the organisation and/or behaviour of the surfactant film, therefore impeding its biophysical function and reducing surface activity. It is also worth considering that surfactant function can be inhibited or inactivated through the presence of other factors such as serum proteins [50], proteases, reactive oxygen and nitrogen species or as a consequence of acute lung injury or inflammation [51,52,53]. While pulmonary inflammation has not yet been investigated there is evidence of neural and systemic inflammation in MPS IIIA mice [54,55].

### 4.2. The Role of Pulmonary Surfactant

Pulmonary surfactant consists predominantly of phospholipids, the most abundant of which is dipalmitoylphosphatidylcholine (DPPC), other unsaturated phosphatidylcholine (PC) species, phosphatidylglycerol, phosphatidylinositol and other minor phospholipids, sphingomyelin and BMP, along with neutral lipids and cholesterol, triglycerides and free fatty acids. Together the phospholipids contribute > 90% of pulmonary surfactant composition, with the remaining approximately 8% consisting of surfactant proteins -A, -B, -C and -D [56]. Surfactant proteins -B and -C combine with phospho- and neutral lipids to provide structural integrity to the interfacial film whilst surfactant proteins -A and -D play an important role in innate immune function [57]. The unique structure of the surfactant lipo-protein film is essential to reduce the work of breathing by maintaining a sufficiently low alveolar surface tension and increasing lung compliance [49]. This lipid-rich mixture is synthesised in the alveolar epithelial type II cell and stored within lamellar bodies, which are modified lysosomes. In response to physical and chemical stimuli, lamellar bodies are exocytosed into the fluid lining of the alveoli, where they swell and unravel to form a membrane-based network from which the multi-layered pulmonary surfactant surface film is derived [58]. Regulation of the interfacial surface tension is achieved as the lipids adsorb to the air-liquid interface to form a surface-active film that lowers the surface tension. Given the biosynthetic origin of lamellar bodies, coupled with evidence of increased accumulation of HS in both tissue and BALF and of disruption to lipid synthesis leading to secondary lipid storage, we propose that there will be downstream effects on the surfactant system that may impact pulmonary surfactant function.

### 4.3. Alterations in Gangliosides, BMP and Cholesterol Composition

Gangliosides are acidic glycosphingolipids that contain one or more sialic acids [59] and are primarily regarded as neuronal material, with previous studies having shown an elevation of the gangliosides GM2 and GM3 in the brain of MPS IIIA mice [60] and very low expression in the healthy adult lung [61]. In mouse neuronal tissue GM3 appears to be co-localised with cholesterol and other glycosphingolipids in specialised lipid rafts [62,63]. For example, GM3 inhibits EGFR-tyrosine kinase to differing degrees depending on its sialic acid species. GM2 was below the level of quantitation in mouse lung tissue in this study, which is consistent with the findings of a recent mouse lung lipidomics study [61]. A significant increase in two of the subspecies of GM3 in MPS IIIA mouse lung tissue was recorded when compared with the control group. As well as being involved in the regulation of numerous cell biological events, including development, trafficking, signalling and cellular interactions, GM3 performs an anti-inflammation role. Studies of GM3 deficient mice reveal an exacerbation in inflammatory pathways in rheumatoid arthritis [63,64]. Gaucher disease is associated with increased insulin resistance and has evidence of increased GM3 concentrations in human plasma, although it is unknown if GM3 has any direct effect on inflammatory pathways in this disease [65,66,67]. There is also evidence of elevated levels of GM3 in mouse cerebellar brain cells in juvenile neuronal ceroid lipofuscinosis but the effect of this has not been determined [65]. There has been no evidence to suggest that GM3 in the lung is affected in either disease. The precise role of GM3 in lung tissue remains largely unknown [61]. 

The specific function of BMP in the lung is also largely unknown. However, in general it plays a significant role in cholesterol distribution [68,69] through the formation of lipid rafts, cell signalling and multivesicular body and small vesicle formation [70,71] in the late endosome-lysosome compartments of the cell. It is a structural isomer of phosphatidylglycerol and is most abundant in the endosome-lysosome system of the cell and generally localised to the membranes of multivesicular bodies [72]. BMP is negatively charged, facilitating the adhesion of soluble positively charged hydrolases and activator proteins, allowing them to degrade substrates at the inner membranes of the lysosome [72]. Thus, BMP has an important role in the regulation of lipid and membrane dynamics in the endosome-lysosome system and an excess of BMP may impair the sorting and dissociation of lysosomal hydrolase complex which affects downstream degradation processes [73]. We have reported for the first time an accumulation of BMP in MPS IIIA lung tissue, and a significant decrease in some BMP species within the alveolar compartment. The implications of the decrease in BMP and its function in surfactant are not entirely known, but our surface activity results suggest that the fluidity of the pulmonary surfactant film may be affected, potentially impairing normal function. 

Cholesterol is an essential component of all cell membranes of all tissues, including lung tissue, and a critical modulator of pulmonary surfactant [72,74]. In combination with surfactant proteins -B and -C, cholesterol is embedded into the lipid bilayers of pulmonary surfactant thereby increasing the viscosity of the membrane [75], enabling pulmonary surfactant to reduce the work of breathing by stabilising alveolar surface tension and increasing lung compliance. Fluidity and viscosity must adjust rapidly in response to inspiration and expiration, relying on the properties of cholesterol to optimise the ordered surfactant film to recover quickly from disruption [75,76,77]. Without cholesterol, the lipid bilayer of pulmonary surfactant is disordered and results in compromised surface tension increasing the work of breathing [76,77]. Cholesterol levels are tightly regulated by transcriptional regulation of cholesterol biosynthesis and cellular uptake as well as by deposition of cholesterol into fat droplets in an esterified form; a type of storage [78]. We have found a significant decrease in total cholesterol in MPS IIIA BALF when compared with control samples. Interestingly, this was driven by a significant decrease in cholesteryl esters (CE) as there was no change in free cholesterol. Cholesteryl esters consist of a free cholesterol unit covalently linked to a long chain fatty acid rendering them more hydrophobic than free cholesterol. They reside in the surface-associated surfactant reservoir to be hydrolysed to enable adsorption of cholesterol to the surface-active layer [79,80]. Therefore, the reduction in CE may reduce the effectiveness of pulmonary surfactant, as well as affecting the availability for recruitment and distribution of free cholesterol into the film. A physiological level of cholesterol is required to promote the optimal level of adsorption of surfactant as both an overabundance and a reduction of cholesterol can inhibit surfactant adsorption [81]. The potential reduction in cholesterol availability suggested by our study may be reflected in the impaired surface activity we observed.

### 4.4. Total Alveolar Phospholipid Pool

For the pulmonary surfactant film to decrease surface tension it must be packed efficiently and maintain interfacial integrity so that during expiration the film is compressed to reduce surface tension to close to 0 mN/m. During inspiration and expansion the surface-active film recruits additional surface-active molecules from the aqueous hypophase [56,82]. The phospholipid pool must have a high turnover to maintain its size and quality as it is decreased by the degradation, oxidation and recycling at the air-liquid interface and refreshed by the secretion of surfactant stored in lamellar bodies [83]. The process of fusion of the lamellar bodies with the cell membrane, exocytosis of surfactant and formation of the surfactant film is a fundamental and lengthy process (seconds to minutes) when compared with endocytic functioning in other cells (milliseconds) [84,85]. Hence, a substantial phospholipid pool size and high turnover are essential to maintain the integrity of the surfactant film during the dynamic breathing process. A significant ~50% reduction in the total phospholipid pool was evident in MPS IIIA BALF when compared to control BALF. The surfactant proteins aid in the regulation of phospholipid pool size. Surfactant protein -A and -C promote the movement of surfactant phospholipids to the surfactant film [86,87], while surfactant proteins -B and -D have significant roles in maintaining surfactant pool sizes, with SP-D deficient mice demonstrating altered surfactant morphology [88,89,90,91,92,93]. The decrease in the phospholipid pool in MPS IIIA mice described here may have a significant impact on the function of pulmonary surfactant. Specifically, the phospholipid pool reduction may be detrimental to film integrity, and there may be further impairments to the film coupled with the reductions of the surfactant proteins compromising the ability to maintain and regulate the pool size.

### 4.5. Surfactant Proteins A-D

Surfactant proteins are synthesised by the alveolar epithelial type II cells where SP-B and -C are transported as they are progressively processed from larger precursors and packaged together with phospholipids from the endoplasmic reticulum to lamellar bodies by way of multivesicular bodies [94] before being secreted into the alveolar space. Surfactant proteins -A and -D are post-translationally modified in the endoplasmic reticulum and Golgi apparatus prior to being constitutively secreted to the hypophase [25,95] to aid in the formation of the highly flexible surfactant film. SP-B in the MPS IIIA mouse was the only surfactant protein maintained at a constant level, which likely reflects SP-B’s vital role in surfactant metabolism and function as evidenced by the fact that an SP-B null mutation is incompatible with life [96]. Conversely, the gene expression of the smaller hydrophobic protein, SP-C was reduced as was the amount of protein in BALF. SP-B and SP-C work closely together to modulate the lipid packing and spreading of the surfactant film to optimise the surface tension of the film during the respiratory cycle. However, while SP-B is critical to life, there is less dependence on SP-C in maintaining surfactant integrity as its levels can fluctuate without compromising function [97]. Functionality of the surfactant film is sustained in part due to SP-C in combination with cholesterol by inducing phase segregation into fluid ordered and fluid disordered sections up to a temperature of 37.5 °C [75,77,98]. SP-C enhances the expression of genes responsible for the metabolism and transport of cholesterol, therefore indirectly playing a role in modulating surfactant fluidity at physiological temperatures [75,99], but SP-C does not affect the phospholipid pool size [100]. Decreased SP-C may also have a deleterious effect on the structure of the septal wall in the ageing lung. Such structural changes are often the precursor to interstitial fibrosis that affects pulmonary mechanics [99,101]. SP-C also contributes to innate host defence in the pulmonary film, enhancing macrophage-mediated phagocytosis and clearance and limiting pulmonary inflammatory responses [99,102]. Deficiencies in SP-C are associated with severe respiratory pathologies [100,103,104,105,106] particularly in interstitial lung diseases in young children [107]. A concomitant reduction in both cholesterol and SP-C at the alveolar surface of MPS IIIA mice has the potential for a significant deleterious effect on the functioning of pulmonary surfactant by increasing the work of breathing and affecting host defence.

The larger surfactant proteins -A and -D are members of the collectin family of proteins and are integral to pulmonary innate immunity by interacting with immune cells, enhancing phagocytosis and killing by macrophages and neutrophils (for review see [35,57,108,109,110]). The reduction in the gene expression of SP-A and -D, the expression of SP-D protein in mouse lung tissue, and the amount of SP-A and -D in BALF, suggest that innate immunity would likely be compromised in the MPS IIIA mouse. As well as their role in innate immunity, SP-A and -D are integral in forming the interfacial surfactant film. Deletion of SP-A affects the adsorption of pulmonary surfactant to the interface and the formation of tubular myelin in the hypophase, hence compromising the integrity of the pulmonary surfactant film [111,112]. Furthermore, SP-D plays a role in regulating the surfactant pool size and ultrastructure of the pulmonary surfactant lining of the lung, with evidence of SP-D deficient mice showing an accumulation of surfactant phospholipids, but no change in surfactant proteins [85,89,90,91]. Interestingly, the alveolar epithelial type II cells of SP-D deficient mice contain giant disorganised lamellar bodies and simultaneously there is an increase in the phospholipid pool size and the appearance and number of foamy alveolar macrophages [89,93]. The re-introduction of SP-D corrects the alveolar macrophage changes [113], strongly suggesting that SP-D plays a significant role in the correct internalisation and recycling of spent surfactant. Hence, SP-A and -D play significant roles in the metabolism, formation of surface-active interfacial film and innate immunity, and therefore the reduced gene and protein expression will likely have a detrimental impact on pulmonary surfactant formation and the innate immune system in the alveolar compartment of the MPS IIIA mouse.

### 4.6. Surfactant Activity

The ability of the surfactant film to adsorb to the air-liquid interface, as it is expanded and compressed during the breathing cycles, is measured in vitro using captive bubble surfactometry (CBS). Pulmonary surfactant, adsorbed at the air-liquid interface, reduces surface tension (γ) from 70 mN/m (at 37 °C) to 20–25 mN/m (equilibrium surface tension, γ_eq_) and then decreases further upon expiration to values of less than 2 mN/m [114]. Under static conditions, surfactant from MPS IIIA mice was able to reach the same equilibrium surface tension upon both initial adsorption and following post-expansion adsorption compared with that from control mice. However, the rate of surface tension reduction was slightly reduced in the post expansion adsorption maneuver, suggesting that there may be inhibitors present in the BALF preventing efficient recruitment and adsorption of new material to the expanding surface in order to rapidly reduce surface tension. Furthermore, upon dynamic cycling surfactant from MPS IIIA mice demonstrated significant increases in both γ_min_ and γ_max_, further supporting the suggestion that surfactant activity was inhibited in the diseased mice. The findings of compromised surfactant activity were further supported by the increase in compressibility (ΔΠ/ΔA) (a low compressibility, i.e., a steeper slope indicates improved efficiency of surfactant to reduce surface tension upon compression), and increased hysteresis (an indirect measure of the work associated with surfactant reorganising itself at the air-liquid interface during compression-expansion cycles). These changes indicate that the surfactant film is not able to pack as tightly on the compression cycle in the MPS IIIA mice when compared to control mice, likely as a consequence of the inclusion of spurious components into the films. Energy efficient surfactant films have a low hysteresis, which is calculated as the area enclosed between the compression and expansion moieties of the isotherms [56]. However, the surfactant from MPS IIIA mice had a higher hysteresis, indicating that it required significantly more energy to decrease the surface tension to a minimum level in the dynamic cycles.

Although surfactant from MPS IIIA mice had reduced surfactant phospholipids and proteins, these factors are unlikely to explain the differences observed in surface activity. We adjusted the quantity of surfactant added to the CBS to maintain a consistent amount of total phospholipid. Because the surfactant proteins were similarly reduced in the MPS IIIA samples the composition of the analysed surfactant is not likely to have differed between control and MPS IIIA samples. Therefore, we propose that the likely contaminating factor that may be inhibiting surface activity is HS or some derivatives thereof, which increased 25-fold in BALF of MPS IIIA mice. The incorporation of macromolecules such as proteins and polysaccharides into the surfactant film has been shown to inhibit surface activity in vitro and in vivo as seen in acute respiratory distress syndrome [53,115,116,117]. However, the specific effect of HS on surface activity is unknown. Given that MPS IIIA patients are characterised by both neuro- and systemic inflammation [118], we cannot discount the possibility that there may also be other contaminating factors such as cytokines, e.g., C-reactive protein and other amine, peptide and lipid inflammatory mediators, which have also been shown to be potent inhibitors of surfactant [50,51]. Finally, the combination of both reduced surfactant quantities and reduced surface activity are likely to have highly detrimental effects on surfactant and lung function in MPS IIIA. Specifically, the higher surface tension and increased hysteresis in MPS IIIA surfactant during dynamic cycling indicate that expansion and compression cycles require more energy, potentially increasing the work of breathing [112,114,119].

## 5. Conclusions

We have described increased storage of HS, GM3 and BMP in MPS IIIA mouse lung tissue. Combined with decreased gene and protein expression of surfactant proteins, this demonstrates altered pulmonary surfactant synthesis in MPS IIIA mice. We have also described an increase in HS as an exogenous contaminant in alveolar surfactant. At the same time there was a decrease in BMP, cholesterol, total phospholipids and surfactant proteins -A, -C and -D in MPS IIIA alveolar surfactant, suggesting impaired secretion. Collectively these changes have contributed to a decrease in surface activity, which is likely to impact lung mechanics. The reduction, particularly in SP-A and SP-D, is also likely to negatively affect pulmonary innate immunity. This mouse model of Sanfilippo syndrome has, for the first time in any mucopolysaccharidosis, shown alterations in pulmonary surfactant contributing to respiratory dysfunction in the distal lung.

## Figures and Tables

**Figure 1 cells-10-00849-f001:**
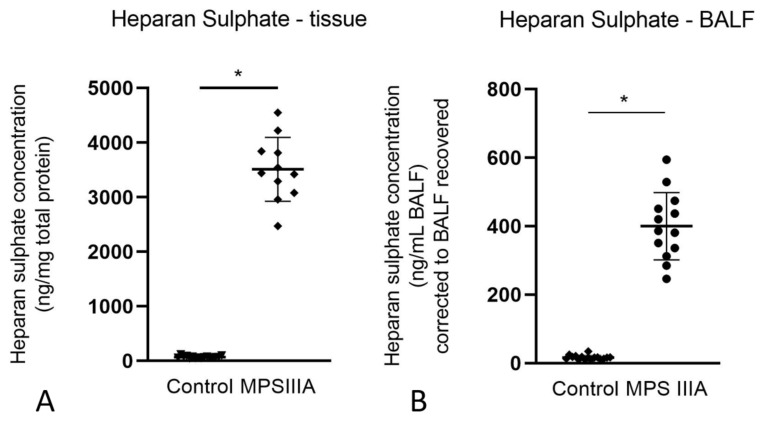
Heparan sulphate analysis in control (i.e., wild type and heterozygous combined) and MPS IIIA mice. HS accumulation in lung homogenates (**A**) and BALF (**B**) of 20-week control and MPS IIIA mice. HS was measured by LC-MS/MS using an API 4000 QTrap mass spectrometer (ABSciex) coupled with Acquity UPLC (Waters Corp). Data for A and B are expressed as disaccharide concentration (ng/mg total protein and ng/mL BALF, respectively) and plotted as mean ± SD; * indicates a significant difference between control and MPS IIIA mice. Two-tailed Students *t*-tests, *p* < 0.0001, *n* = 17 control and 11 MPS IIIA samples for lung tissue, *n* = 16 control and 13 MPS IIIA samples for BALF.

**Figure 2 cells-10-00849-f002:**
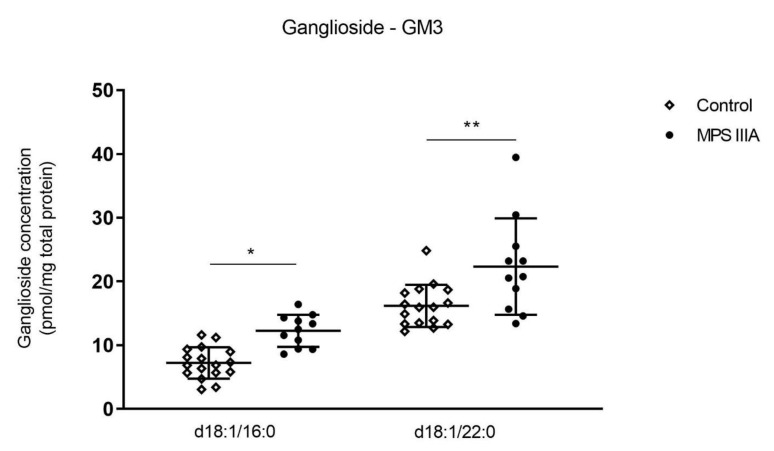
Ganglioside analysis. Ganglioside GM3 concentration in lung homogenates of 20-week control and MPS IIIA mice. Ganglioside GM3 was measured by LC MS/MS using an API 4000 QTrap mass spectrometer (ABSciex) coupled with a Prominence HPLC (Shimadzu). Data are expressed as ganglioside concentration (pmol/mg total protein) to the GM standard (GM1 (d35) d18:1/18:0) and plotted as mean ± SD; two-tailed Students *t*-tests, * and ** indicate a significant difference between control and MPS IIIA mice, *p* < 0.0001 and 0.0077, respectively, *n* = 17 control and 11 MPS IIIA samples.

**Figure 3 cells-10-00849-f003:**
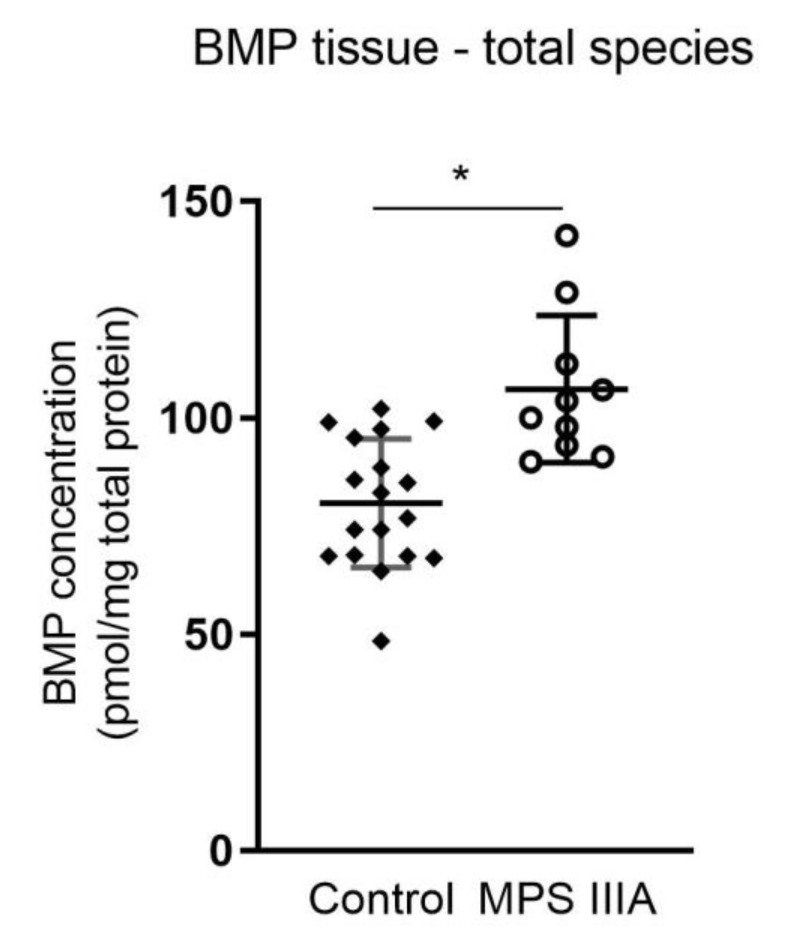
Total bis(monoacylglycerol)phosphate (BMP) analysis. Total BMP concentration in lung homogenates of 20-week control and MPS IIIA mice. Quantification of the BMP molecular species was measured by LC MS/MS using an API 4000 QTrap mass spectrometer (ABSciex) coupled with a Prominence HPLC (Shimadzu). Data are expressed as BMP concentration (pmol/mg total protein) to the BMP standard (BMP 14:0/14:0(Avanti 840445P)) and plotted as mean ± SD; two-tailed Students *t*-test * indicates a significant difference, *p* = 0.0002, two-tailed Students *t*-test; *n* = 18 control and 10 MPS IIIA samples.

**Figure 4 cells-10-00849-f004:**
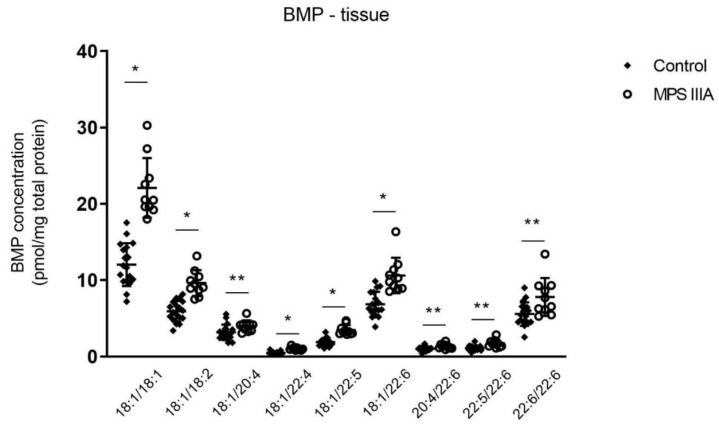
Bis(monoacylglycerol)phosphate (BMP) molecular species analysis. BMP molecular species concentration in lung homogenates of 20-week control and MPS IIIA mice. Quantification of the BMP molecular species was measured by LC MS/MS using an API 4000 QTrap mass spectrometer (ABSciex) coupled with a Prominence HPLC (Shimadzu). Data are expressed as BMP concentration (pmol/mg total protein) to the BMP standard (BMP 14:0/14:0(Avanti 840445P)) and plotted as mean ± SD; two-tailed Students *t*-tests, * and ** indicate a significant difference, *p* < 0.0001 and <0.05, respectively between control and MPS IIIA mice; *n* = 18 control and 10 MPS IIIA samples.

**Figure 5 cells-10-00849-f005:**
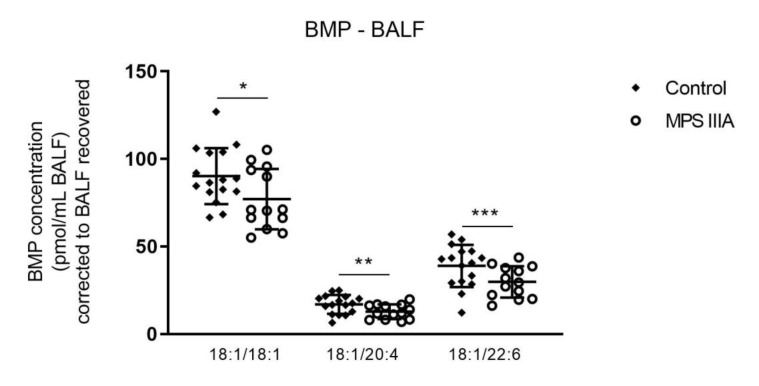
Bis(monoacylglycerol)phosphate (BMP) analysis of BALF. Quantification of the BMP molecular species in 20-week control and MPS IIIA BALF. BMP was measured by LC MS/MS using an API 4000 QTrap mass spectrometer (ABSciex) coupled with a Prominence HPLC (Shimadzu). Data are expressed as BMP concentration (pmol/mL BALF) to the BMP standard (BMP 14:0/14:0 (Avanti 840445P)) and plotted as mean ± SD; two-tailed Students *t*-tests, *, ** and ***indicate a significant difference, *p* = 0.0430, 0.0300 and 0.0325, respectively between control and MPS IIIA mice; *n* = 17 control and 11 MPS IIIA samples.

**Figure 6 cells-10-00849-f006:**
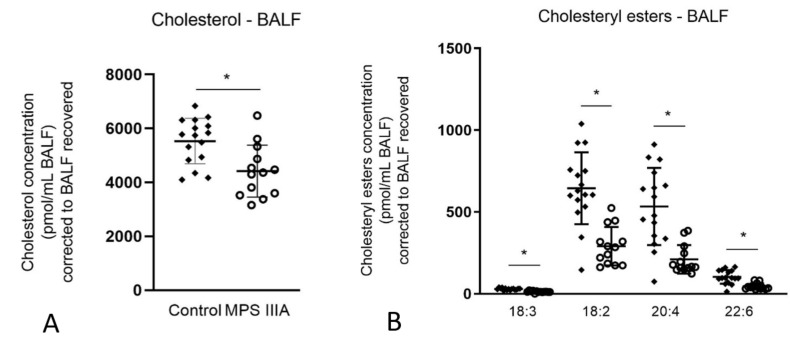
Cholesterol analysis of BALF (**A**) and cholesteryl ester analysis of BALF (**B**). Quantification of cholesterol and the ester molecular species in 20-week control and MPS IIIA BALF were measured by LC MS/MS using an API 4000 QTrap mass spectrometer (ABSciex) coupled with a Prominence HPLC (Shimadzu). Data are expressed as concentration (pmol/mL BALF) to C17 Cholesteryl ester (Sigma C5384) and plotted as mean ± SD; two-tailed Students *t*-test, * indicates a significant difference, *p* < 0.0001 between control and MPS IIIA mice, *n* = 16 control and 13 MPS IIIA samples (**A**) and *n* = 15 control and 13 MPS IIIA samples (**B**).

**Figure 7 cells-10-00849-f007:**
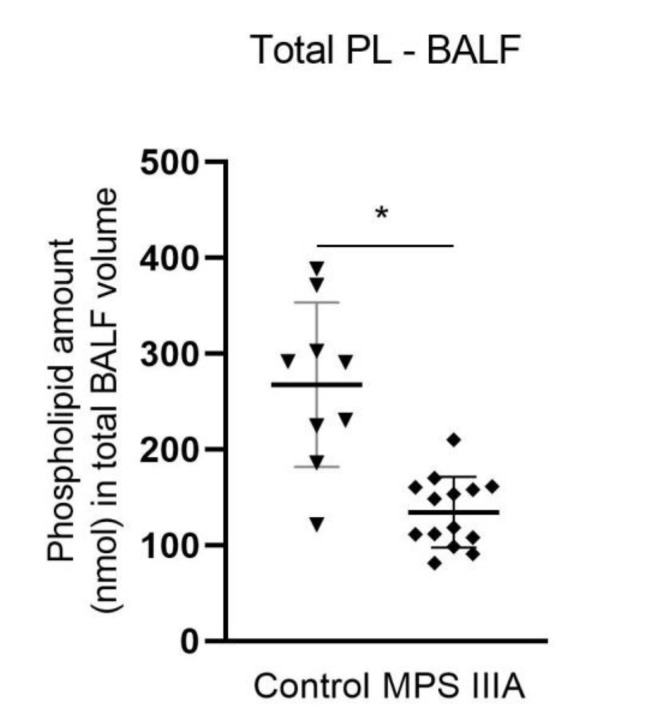
Total phospholipid pool size analysis in BALF. Quantification of the phospholipid species in control and MPS IIIA BALF. Phospholipids were measured using electrospray ionisation mass spectrometry (ESI-MS) (Waters XEVO TQ-MS). Data are expressed as total phospholipids (nmol) in the total corrected BALF volume and plotted as mean ± SD; two-tailed Students *t*-test, * indicates a significant difference, *p* < 0.0001 between control and MPS IIIA mice; *n* = 9 control and 14 MPS IIIA samples.

**Figure 8 cells-10-00849-f008:**
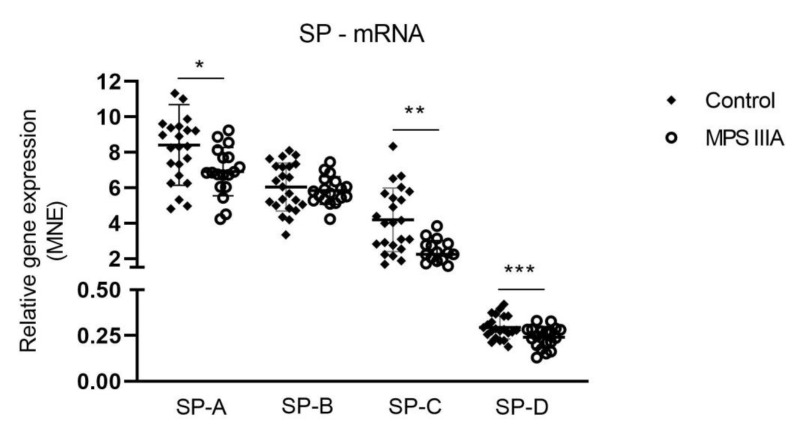
Mean normalised expression (MNE) of surfactant proteins -A, -B, -C and -D mRNA. Relative gene expression of surfactant proteins in control and MPS IIIA lung tissue. Data expressed as MNE and plotted as mean ± SD; two-tailed Students *t*-tests; *, ** and *** indicate significant differences *p* = 0.0184, *p* = 0.0001 and *p* = 0.0073, respectively between control and MPS IIIA mice; *n* = 23 control and 18–19 MPS IIIA samples.

**Figure 9 cells-10-00849-f009:**
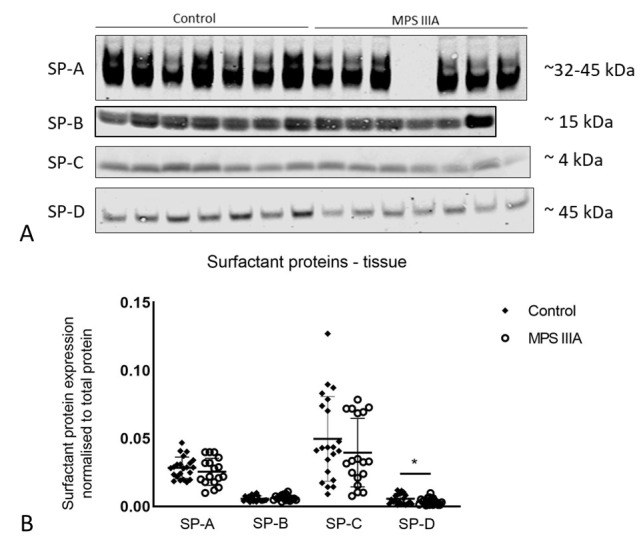
Protein expression of surfactant proteins -A, -B, -C and -D. (**A**) Representative Western blot images of part sets of control and MPS IIIA lung tissue samples probed for SP-A (primary antibody-rabbit Thermo Scientific PA5-76699), SP-B (primary antibody-rabbit SevenHills Bioreagents WRAB-48604), SP-C (primary antibody-rabbit SevenHills Bioreagents WRAB-MSPC) and SP-D (primary antibody-rabbit Chemicon AB3434). (**B**) Data normalised to total protein and presented as mean ± SD; two tailed Students *t*-tests, * indicates a significant difference, *p* = 0.0443 between control and MPS IIIA mice; *n* = 22 control, 17 MPS IIIA (SP-A analysis); 20 control, 16 MPS IIIA (SP-B analysis); 21 control, 18 MPS IIIA (SP-C analysis) and 22 control, 19 MPS IIIA (SP-D analysis) samples. Entire band height of SP-A measured from 32–45 kDa [35].

**Figure 10 cells-10-00849-f010:**
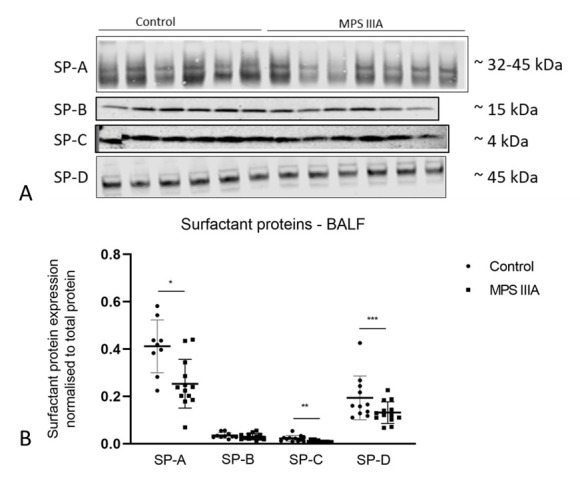
Protein expression of surfactant proteins -A, -B, -C and -D in BALF. (**A**) Representative Western blot images of part sets of SP-A, -C and -D samples pictured. Western blot analyses were performed with the following primary antibodies: SP-A (rabbit Thermo Scientific PA5-76695), SP-B (rabbit SevenHills Bioreagents WRAB-48604), SP-C (rabbit SevenHills Bioreagents WRAB-MSPC) and SP-D (rabbit Thermo Scientific PA5-51476). (**B**) Data normalised to total protein and presented as mean ± SD; significant differences are indicated * *p* = 0.0027, ** *p* = 0.0024 and *** *p* = 0.0438; two tailed Students *t*-test; *n* = 9 control, 13 MPS IIIA (SP-A analysis); 10 control, 13 MPS IIIA (SP-B analysis); 12 control, 13 MPS IIIA (SP-C analysis) and 11 control, 13 MPS IIIA (SP-D analysis) samples. Entire band height of SP-A measured from 32–45 kDa [35].

**Figure 11 cells-10-00849-f011:**
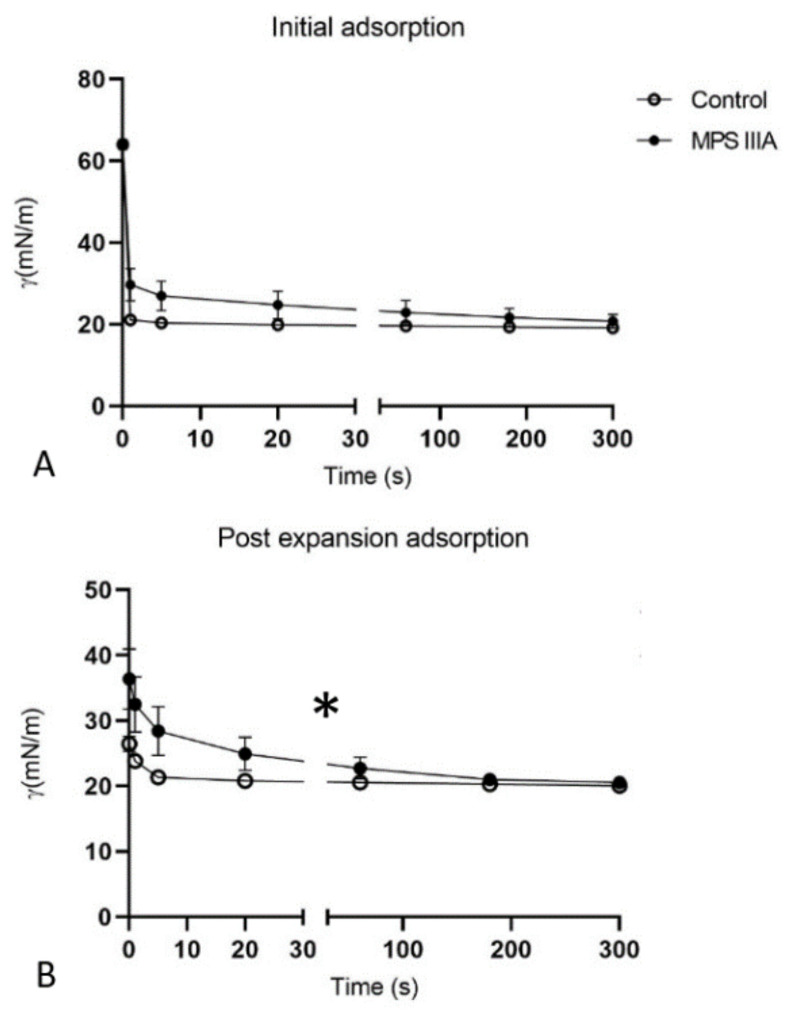
Captive bubble surfactometry of BALF samples under static conditions. Surfactant biophysical properties of control and MPS IIIA BALF under static conditions—initial adsorption (**A**) and post expansion adsorption (**B**). Data are expressed as γ (m/Nm) and plotted as mean ± SD; repeated measures ANOVA (**A**) *p* < 0.0001 (time effect); (**B**) * *p* = 0.0253 (disease effect), *p* < 0.0001(time effect) and there was an interaction between genotype and time; *n* = 10 control and 6 MPS IIIA samples.

**Figure 12 cells-10-00849-f012:**
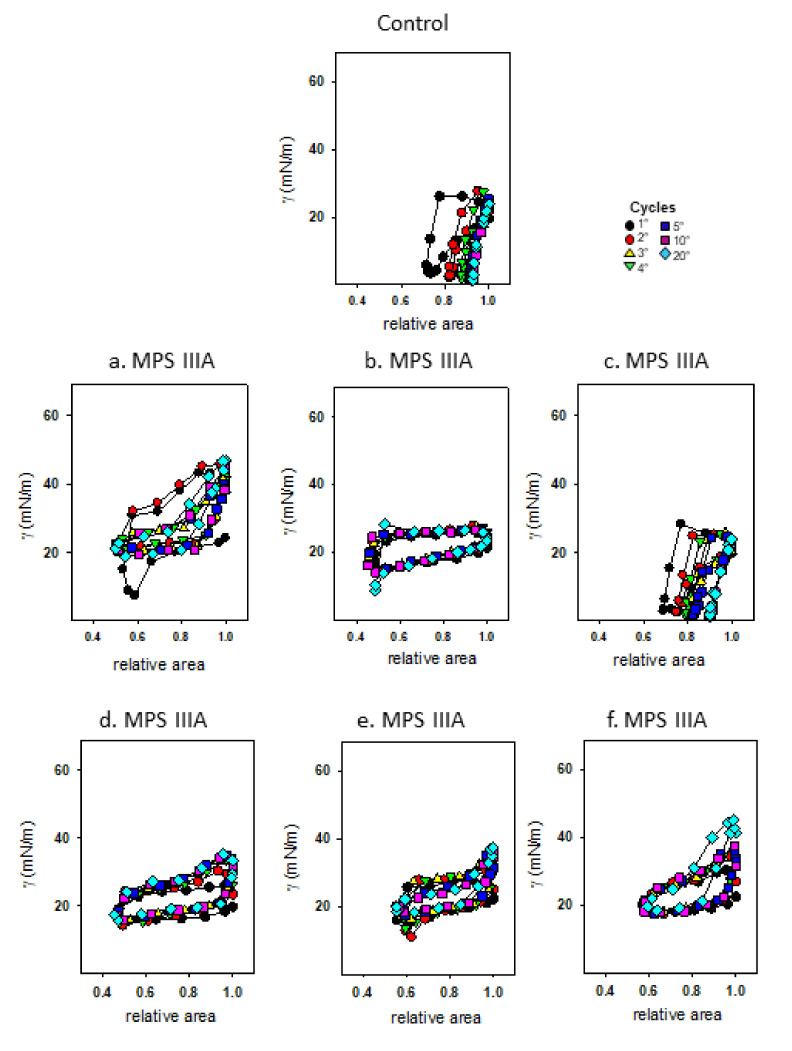
Captive bubble surfactometry of BALF surfactant samples under dynamic conditions. Biophysical properties of representative control and each individual MPS IIIA surfactant sample subjected to dynamic compression-expansion cycling which mimics spontaneous respiratory cycles. The lower limb of the curve represents the compression of the bubble, and the upper limb of the curve was obtained upon expansion. Thirty cycles were performed at 30 cycles per min with cycles 1, 2, 3, 4, 5, 10 and 20 shown.

**Figure 13 cells-10-00849-f013:**
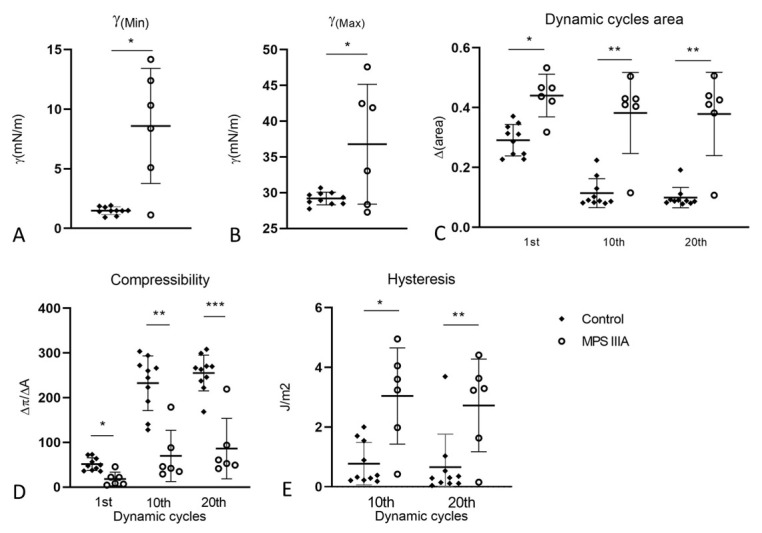
Biophysical properties of surfactant from BALF under dynamic compression cycling. Minimum (**A**) and maximum (**B**) surface tension of control and MPS IIIA BALF surfactant samples expressed as mN/m. MPS IIIA animals have higher minimum surface tension, *p* < 0.003 (**A**), and higher maximum surface tension, *p* < 0.0115 (**B**). (**C**) Relative area of compression required to reach γ_min_ was calculated as the percentage of area reduction needed to decrease the surface tension to the minimum value. Change in relative area in the 1st, 10th and 20th cycles was increased in MPS IIIA samples, * *p* = 0.0003, ** *p* < 0.0001. (**D**) Compressibility (ΔP/ΔA) was calculated as the slope value of the line passing through the minimum and maximum points of compression (minimum and maximum areas of the bubble) in the dynamic cycling curve (the steeper the slope, i.e., the higher the value, the lower the surfactant compressibility). * *p* = 0.0006, ** *p* = 0.0001, *** *p* < 0.0001. (**E**) Hysteresis was calculated as the enclosed area between the compression and expansion segments of the isotherms. * *p* = 0.0015, ** *p* = 0.0076. Data plotted as mean ± SD; two-tailed Students *t*-tests used for all analyses; *n* = 10 control and 6 MPS IIIA samples.

**Table 1 cells-10-00849-t001:** Chromatographic gradient used for the analysis of butanolic products of HS.

Time	Solvent A (%)	Solvent B (%)	Flow (µL/min)
**0.01**	99	1	0.350
**2.00**	95	5	0.350
**2.01**	80	20	0.350
**5.00**	75	25	0.350
**5.01**	1	99	0.350
**7.50**	99	1	0.350
**8.50**	99	1	0.350

**Table 2 cells-10-00849-t002:** Internal standard composition 100 µL per sample.

Standard Name	nmol Added to Each Sample at Extraction
Dimyristoyl-phosphatidylcholine (DMPC)	10
Dimyristoyl phosphatidylethanolamine (DMPE)	4
Dimyristoyl phosphatidylserine (DMPS)	2
Dimyristoyl phosphatidylglycerol (DMPG)	2
Dimyristoyl phosphatidic acid (DMPA)	1
Lysophosphatidylcholine (LPC17:0)	1
Sphingomyelin (SM16:0)	1
Ceramide 12:0	2
Triacylglycerol (TAG)	10

**Table 3 cells-10-00849-t003:** ESI Diagnostic MS/MS scans.

MS	Scan	MS/MS Mode	Range (*m*/*z*)
ES+	a	Full positive	2–1300
ES-	a	Full Negative	2–1300
ES-	a	Partial Negative	400–900

a = first injection.

**Table 4 cells-10-00849-t004:** qRT-PCR primer sequences and optimal primer concentrations for housekeeping genes and genes of interest.

Gene	Primer Sequence (5′ → 3′)	Accession Numbers
ACTB	FWD-AGCTGTGCTATGTTGCTC REV-CACTTCATGATGGAATTGAATGTAG	BC138614.1
HPRT1	FWD-GCTGGATTACATTAAAGCACTGAAT REV-AAAGTTTGCATTGTTTTACCAGTGT	NM_013556
YWHAZ	FWD-TTGAGCAGAAGACGGAAGGT REV-GAAGCATTGGGGATCAAGAA	NM_011740
PGK1	FWD-GTCGTGATGAGGGTGGACTT REV-TTTGATGCTTGGAACAGCAG	NM_008828
RpPo1	FWD-CAACCCTGAAGTGCTTGACAT REV-AGGCAGATGGATCAGCCA	NM_007475.5
PPIA	FWD-ACCAAACACAAACGGTTCCC REV-TGCCTTCTTTCACCTTCCCAAA	NM_008907.1
SP-A	FWD-TTTCCACCAATGGGCAGTCA REV-AGAAGCCCCATCCAGGTAGT	NM_023134
SP-B	FWD-GCTACTGCTGCTTCCTACCC REV-TGGCACAGGTCATTAGCTCC	NM_147779
SP-C	FWD-GGAGCACCGGAAACTCAGAA REV-GGAGCCGCTGGTAGTCATAC	NM_011359
SP-D	FWD-AGCCCAACAACAATGGTGGA REV-CACAGATAACAAGGCGCTGC	NM_009160

ACTB, β-actin; HPRT1, hypoxanthine phosphoribosyltransferase 1; YWHAZ; tyrosine 3-mono-oxygenase/tryptophan 5-mono-oxygenase activation protein; PGK1, phosphoglycerate kinase 1; RpP0, ribosomal protein P0 and PPIA, peptidylprolyl isomerase A. FWD, forward primer; REV, reverse primer. SP-A, surfactant protein-A; SP-B, surfactant protein-B; SP-C, surfactant protein-C; SP-D, surfactant protein-D. FWD, forward primer; REV, reverse primer.

## Data Availability

The data presented in this study are available in this article and the associated Appendix A.

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
