# Peer review of "Increased Alveolar Heparan Sulphate and Reduced Pulmonary Surfactant Amount and Function in the Mucopolysaccharidosis IIIA Mouse"

_cells, 2021, doi:10.3390/cells10040849_

Round 1
Reviewer 1 Report
The manuscript reports the results from a research project aiming at delineating the alterations in pulmonary tissue storage of glycosaminoglycan (GAG), heparan sulphate (HS), lipid and protein composition of lung tissue and pulmonary surfactant, and surfactant activity using a naturally occurring MPS IIIA (Sanfilippo syndrome) mouse model. The hypothesis was that impaired surfactant metabolism and activity might explain the increased susceptibility of MPS IIIA patients to respiratory infections and other pulmonary insults.
The aim of the study was clear and straightforward and the Materials and Methods were clearly described, well-designed and nicely executed. The study group reported an increase in HS in both lung tissue and alveolar surfactant of MPS IIIA mice compared with unaffected control mice. Increases in ganglioside GM3 and BMP lipid species, decreases in surfactant proteins -A, -C and -D mRNA expression and a decrease in SP-D protein in lung tissue were also found. It is a very important finding to demonstrate an impairment of surfactant composition and function in the mucopolysaccharidoses. The group also reported for the first time a significant increase in the content of HS in the bronchoalveolar lavage fluid (BALF) of MPS IIIA mice. Study using captive bubble surfactometry on the BALF samples subjected to dynamic compression-expansion cycling showed damaging biophysical changes in MPS IIIA alveolar surfactant. All these abnormal changes in alveolar surfactant are likely to be detrimental to lung function in MPS IIIA. As most of the literature described minimal manifestations of restrictive lung disease, lower airway obstruction and upper airway obstruction in individuals with MPS III among all the MPS disorders, this report provides us quite persuasive perspectives on how we should revisit and re-evaluate the way we manage the respiratory problems in MPS III.
However, the respiratory disorders of MPS III, and also other MPS disorders, evolved from a very dynamic pathogenic changes in the whole respiratory system with significant neurologic influences. In the study, the airway movements during active respiration were not evaluated and documented. The relaxation and contraction of the airway smooth muscle, together with the structure and composition of the tracheobronchial tree, influence the airway lumen and resistance. The evaluation of the damaging effects from HS accumulation in the respiratory system should include histopathological examination and immunologic assessment. Nonetheless, as the authors have claimed, the study on MPS IIIA mouse model has, for the first time in any MPS, shown alterations in pulmonary surfactant contributing to respiratory dysfunction in the distal lung. The fruitful results from the research project might shine a light on why patients with MPS IIIA experience reduced lung function and repeated lower respiratory tract infections.
Author Response
As indicated in the introduction the scope of this paper is on pulmonary surfactant and any alterations in the production of the lipid and protein components and amounts. Changes in these components have been observed in other lysosomal storage disorders, therefore it was our intent that this be a starting point for the investigation in the mucopolysaccharidosis model available to us. We understand the reviewer’s comments regarding investigation of lung function, histopathological and immunologic assessments and wish to assure the reviewer that these elements, in addition to stereological evaluation using light and electron microscopy, are part of the further investigation that is now warranted in the broader research project.
Reviewer 2 Report
Paget et al. showed the biochemical changes such as heparan sulphate, GM3 ganglioside and BMP in lung and BALF from MPS IIIA mice. They also indicated biophysical change in surfactant in MPS IIIA mice. Although data are interesting, I have several concerns.
Concerns
1 Authors indicated the accumulation of heparan sulphate in lung homogenates and BALF from MPS IIIA mice. However, sulphamidase activity in lung homogenate and BALF is not shown in this manuscript. Please show the enzymatic activity of SGSH in lung and BALF from MPSIIIA mice.
2 Authors combined quantitative data from WT and hetero mice as control in this manuscript. Although authors showed the similar levels of heparan sulphate in WT with that in hetero mice, the levels of other analytes are not indicated. Please show the actual data of all analytes from WT and hetero mice, respectively.
3 Whereas statistically difference was described in all figures, error bars are seems to be large in several figures such as GM3 ganglioside , BMP and cholesterol ester. Please indicate the actual numbers (mean +/- SEM ) about all analysis. In addition, I recommend the replace the bar graph to scatter plot with bar graph in all figures.
Author Response
- Enzymatic activity in this model of the MPS IIIA mouse has been reported to be 3-4% as measured in the liver (Crawley et al., 2006). In addition, the mouse model from which the Crawley model has been developed has also shown liver, kidney and brain enzyme activity at between 3-4% (Bhaumik et al., 1999). Subsequently, given that the enzyme activity in organs has been consistently established between 3 and 4% and we have shown an increase in heparan sulphate in homogenate (50-fold) and BALF (25-fold) which supports low enzymatic activity, we feel that the reviewer’s request to provide these data is not warranted. Changes have been made to explain the model in lines 91-93.
- We respect the view of Reviewer 1 to provide the data from WT and heterozygous groups. The individual values for WT and het animals for all analytes have been included in Supplementary Material – Table C (line 881). However, we do respectfully draw Reviewer 1’s attention to the comments of Reviewer 2, who has suggested that we remove the graphical explanation of the Heparan sulphate data with regards to WT and het mice (original Fig 1) and that the data with a comment in the text is sufficient.
- We have left all graphical representation as control and MPS IIIA but have changed the graphs to scatter plots per Reviewer 1 recommendation and mean ± SD as per Reviewer 2 recommendation. (All figures)
Crawley, A. C., Gliddon, B. L., Auclair, D., Brodie, S. L., Hirte, C., King, B. M., Fuller, M., Hemsley, K. M., & Hopwood, J. J. (2006). Characterization of a C57BL/6 congenic mouse strain of mucopolysaccharidosis type IIIA. Brain Research, 1104(1), 1-17.
Bhaumik, M., Muller, V. J., Rozaklis, T., Johnson, L., Dobrenis, K., Bhattacharyya, R., Wurzelmann, S., Finamore, P., Hopwood, J. J., Walkley, S. U., & Stanley, P. (1999). A mouse model for mucopolysaccharidosis type III A (Sanfilippo syndrome). Glycobiology, 9(12), 1389-1396.
Reviewer 3 Report
In this study, Paget et al. compared the amount of Heparan sulphate, gangliosides, cholesterol, and phospholipids in lung and BALF and the characteristics of surfactant between wildtype and MPS IIIA mice in order to get a better understanding on the lung problems that are commonly found in MPS patients. They observe a clear increase of HS amount in lung and BALF and some rather small changes in gangliosides, cholesterol esters, and phospholipids. They also characterized the biophysical properties of surfactant and could observe clear differences here and conclude that alterations in pulmonary surfactant contribute to the respiratory complications in MPSIII.
The approach is interesting and relevant, because respiratory complications occur in man lysosomal storage diseases and contribute significantly to the morbidity, especially in adult patients. Recurrent airway infections are frequent in many lysosomal storage diseases and are not restricted to the MPS group of diseases.
However, the manuscript needs to be improved. It is way too long and contains too many Figures and Tables, many of which don’t contain relevant information. Especially the methods, results and discussion need to be shortened and be made more concise. Currently, the manuscript resembles a thesis rather than a journal article.
Specific comments:
- The introduction is fine.
- Materials and Methods need be shortened a lot. Tables 4 and 5 can be omitted or transferred to the Supplementary section. Combine Tables 6 and 7. If the final primer concentration is the same for all primers, the column can be deleted in Table 6/7. Table 8 is not necessary, antibodies can be listed in the text.
- Results:
- Figure 1 can be deleted. Just mention that WT and HET were combined.
- Why was SEM used instead of SD? To show how big the differences between the groups are, SD should be used instead of SEM in all Figures.
- Table 9 can be deleted, just mention that there was no difference.
- Are Tables 10 and 11 necessary? If yes, transfer to Supplementary section.
- Table 12 can be deleted and Figure 7 also is not necessary or can be combined with Figure 8.
- Figures 9-11: The phospholipid analysis did not show differences between genotypes. The indicated significant changes to not seem to be very big. It is enough to mention this in the text.
- Figure 12: The small changes in mRNA expression, especially those of SP-A and SP-D are not convincing to me.
- Figure 13: Why do the blot images contain different number of samples? SP-A exposure is too dark, no interpretation is possible.
- Figure 14: SP-C looks photoshopped on the left side. From the blot images, I can’t see significant differences in the amount of any of the 4 proteins.
- The discussion needs to shortened. Section 4.5 should be changed because the data in the manuscript do not support a clear role of surfactant proteins.
Author Response
- Thank you for the positive commentary re the introduction.
- We appreciate the reviewer’s remarks with regards to tables 4-8 and have adjusted as suggested. Table 4 and 5 have been moved to the Supplementary Material and labelled Table A (line 875) and B (line 879). Tables 6 and 7 have been combined and labelled Table 4 (line 327). Table 8 has been included in text (lines 374-382).
- We have provided a comment in the text re combining WT and heterozygous data to form a control group of mice (line 404-409) and also provided a table of WT, het, combined control and MPS IIIA data for all analytes in the supplementary section. We draw the attention of reviewer 2 to the comments of reviewer 1 (point 2) with regards to the inclusion of WT and het data and point 3 with regard to the inclusion of all mean ± SD data.
- We have changed the graphical representation to SD rather than SEM, and have changed all graphs to scatter plots in accordance with reviewer 1’s request. All graphs.
- Data from Table 9 have been included in Supplementary Material – Table C (line 881).
- Data from Tables 10 and 11 have been included in Supplementary Material – Table C (line 881).
- Data from Table 12 have been included in Supplementary Material – Table C (line 881). Figure 6 and 7 have been combined and labelled Figure 6 (A and B) (line 481).
- Figures 9 and 10 have been deleted and the data recorded in the Supplementary Material - Table C (line 881). Figure 11 has remained and labelled Figure 7 (line 504).
- Thank you for your comments re the mRNA data. In response to Reviewer 1’s request for more complete data we have included data in Supplementary Material – Table C. What can be determined from these data, and also the graphs is that the actual decreases in the mean are approximately 19%, 38% and 20% respectively. Therefore, whilst considered modest, they are not negligible and should be regarded as significant changes in accordance with the outcomes of the statistical analysis. Moreover, their biological significance is reinforced by consistent changes across a range of surfactant compositional and functional parameters.
- We thank Reviewer 2 for drawing our attention to this detail, as it should have been included in the first instance. Re different number of samples – sample size of lung homogenates is too big to be contained on one gel, rather we used three gels for all of the samples; sample size of BALF required two gels. Each gel contained control and MPS IIIA samples. For consistency, all samples were tested in triplicate, included in the methods at (line 384). As a representative blot was used for illustrative purposes in each instance, it is therefore not feasible to determine quantitative differences in protein based exclusively on the visual analysis of a representative blot.
Re SP-A exposure – SP-A is varibly glycosylated which produces bands from 32kDa to 45kDa. Therefore we have measured the entire area between ~32-45kDa (Vieira et al., 2017). We have amended the methods to explain that samples were tested in triplicate (line 384), and provided a reference to the SP-A band width sampling. (lines 533 and 542)
- TIFF files of the original gel images were submitted with the original manuscript. We can absolutely assure Reviewer 2 that none of the images were photoshopped. The uneveness of the SP-C gel is where the gel had torn slightly in handling prior to transfer to the membrane.
With respect to quantification the procedure for western blot analysis normally assumes that the same amount of protein, as determined by a BCA, is loaded into each lane. However, this also leaves room for error in both the BCA analysis and loading. For a consistent quantitative analysis we have therefore measured the proteins of interest and total protein (using Revert) in each lane using the Image Studio Lite program, and expressed these as a percentage thus removing the potential procedural errors (lines 384-385).
- We thank the reviewer for his opinion of the western blots. We believe that the above explanation is adequate to support the changes described, although modest, in the surfactant proteins so have not altered the discussion.
Vieira, F., Kung, J. W., & Bhatti, F. (2017). Structure, genetics and function of the pulmonary associated surfactant proteins A and D: The extra-pulmonary role of these C type lectins. Annals of Anatomy - Anatomischer Anzeiger, 211, 184-201.
Round 2
Reviewer 2 Report
The paper was well revised by authors.
Reviewer 3 Report
The manuscript has been clearly improved.